# Gingival Crevicular Fluid Cytokines in Moderate and Deep Sites of Stage III Periodontitis Patients in Different Rates of Clinical Progression

**DOI:** 10.3390/biomedicines8110515

**Published:** 2020-11-18

**Authors:** Federica Romano, Wilma Del Buono, Laura Bianco, Martina Arena, Giulia Maria Mariani, Federica Di Scipio, Giovanni Nicolao Berta, Mario Aimetti

**Affiliations:** 1Department of Surgical Sciences, C.I.R. Dental School, Section of Periodontology, University of Turin, 10126 Turin, Italy; federica.romano@unito.it (F.R.); wilmadelbuono@gmail.com (W.D.B.); laura.lb.bianco@gmail.com (L.B.); martinarena.mrt@gmail.com (M.A.); giuliamaria.mariani@unito.it (G.M.M.); 2Department of Clinical and Biological Sciences, University of Turin, 10043 Orbassano, Italy; federica.discipio@unito.it

**Keywords:** angiogenesis, gingival crevicular fluid, inflammation mediators, periodontal/therapy, periodontitis

## Abstract

Clinical criteria are inappropriate to measure the degree of susceptibility to progression of periodontal damage. Thus, the aim of this study was to assess whether gingival crevicular fluid (GCF) levels of cytokines could discriminate patients suffering from stage III periodontitis with moderate (Grade B) and rapid rates of progression (Grade C) prior to and 6 months after non-surgical periodontal treatment. GCF samples were obtained from moderate and deep sites of 20 patients diagnosed as Grade B and 20 patients as grade C stage III periodontitis and analyzed for interleukin (IL)-1β, IL-9, tumor necrosis factor (TNF)-α, and vascular endothelial growth factor (VEGF) using a high-sensitivity Bio-Plex Suspension Array System. At baseline, higher IL-1β but lower IL-9 GCF levels were observed in moderate sites of the grade C compared to the grade B group. In spite of comparable clinical improvement, this difference maintained after treatment, suggesting a residual pro-inflammatory state. In deep sites, no differences were observed between periodontitis groups except for VEGF levels that decreased more in Grade B periodontitis at 6 months post-therapy. A mathematical model was constructed to identify Grade C periodontitis patients based on the subjects’ GCF levels of IL-1β and IL-9, which achieved an area under the receiver-operating characteristic (ROC) curve of 0.94. This study can contribute to the early assessment of risk of future breakdown in periodontitis patients.

## 1. Introduction

Periodontitis is a biofilm-induced chronic inflammatory disease involving a complex interplay between immune/inflammatory pathways and symbiotic ecological changes, leading to loss of periodontal attachment [1]. The susceptibility to periodontitis appears to be determined by the host response, specifically, by the magnitude of the inflammatory challenge and the differential activation of immune pathways [2]. The magnitude and consistency of immune-inflammatory response are controlled at a number of regulatory, genetic and epigenetic levels that likely influence the individual variability in the severity and rate of progression of the disease. Consistently, experimental gingivitis studies have revealed that the intensity of the gingival inflammation in periodontitis-susceptible patients is strictly related to the individual inflammatory host response rather than being entirely due to differences in the amount and composition of the bacterial biofilm [3,4,5,6].

The current classification of periodontal diseases has characterized individual cases of periodontitis by staging and grading in order to capture the degree of periodontal breakdown and the risk of disease progression. This is based on direct or indirect evidence of longitudinal loss of attachment or progressive alveolar bone destruction [7]. As periodontitis manifestations result from prolonged and/or exaggerated host inflammatory reactions to a subgingival dysbiotic microbiome, it can be speculated that the periodontal levels of pro-inflammatory mediators may be higher in patients with faster rates of progression [8]. It can also be hypothesized that patients with higher susceptibility to disease progression may respond less favorably to non-surgical periodontal therapy regardless of periodontitis severity. In this context periodontitis is now regarded as a chronic non-resolving inflammatory disease [9].

A non-invasive technique to check the local host response includes the evaluation of biomarkers in the gingival crevicular fluid (GCF) [10,11]. Evidence regarding the GCF levels of mediators modulating angiogenesis and inflammatory/immune response before and after periodontal treatment is limited to patients suffering from chronic and aggressive periodontitis according to the 1999 classification system [12,13,14,15,16]. Angiogenesis is an important aspect of inflammation and healing, but its role in the progression of periodontal lesions is not yet completely clarified [17]. Vascular endothelial growth factor (VEGF) is a multifunctional cytokine inducing proliferation of endothelial cells and increasing vascular permeability [18]. In chronic and aggressive periodontitis subjects, its production is unregulated in diseased sites if compared to healthy sites [19,20], raises proportionally with the severity of the periodontal tissue destruction [21], and is reduced after non-surgical periodontal therapy [15,22]. VEGF expression is regulated by IL-1 and TNF-α [23,24], which are the major mediators in the early inflammation process, able to induce connective tissue demolition and resorption of alveolar bone [25], but also involved in the immune response of neutrophil and T-helper type 1 (Th1) cells [26]. Lately, it has been shown that Th9 cells interact with the Th1 subpopulation in the modulation of inflammatory/immune responses by releasing IL-9 [27]. In different studies, it has been postulated that this cytokine exerts an anti-inflammatory effect by regulating IL-1 and TNF-α production in human granulomas and GCF of subjects with chronic periodontitis [28,29].

There is still a lack of information on GCF levels of cytokines involved in angiogenesis and inflammatory mechanisms in slowly and highly progressive destructive periodontal disease in accordance with the current classification. In addition, another aspect of novelty of our work is data stratification according to the initial probing depth (PD). This information could improve patient monitoring and control of the disease process before extensive periodontal damage has occurred. Clinically, at least 2 mm of clinical attachment loss must occur before a site is deemed as rapidly progressing [7]. Thus, the early identification of sites demonstrating high susceptibility for disease progression would assist the establishment of specific supportive treatment strategies resulting in more predictable treatment outcomes and cost savings. 

Therefore, the primary aim of this preliminary study was to investigate GCF levels of IL-1β, IL-9, TNF-α, and VEGF before and after non-surgical periodontal treatment in patients suffering from stage III periodontitis with moderate and rapid rates of progression. The secondary aim was to determine whether GCF cytokine levels could be useful as a biosignature of disease progression in moderate and deep pocket sites of stage III periodontitis.

## 2. Experimental Section

### 2.1. Study Design 

Subjects participating in the study were serially recruited from the patients referred to the Section of Periodontology, C.I.R. Dental School, Department of Surgical Sciences, University of Turin (Italy) between September 2018 and January 2020. The research was approved by the Institutional Ethics Committee of the “AOU Città della Salute e della Scienza”, Turin, Italy (No. 0027219) and was conducted according to the Helsinki Declaration of 1975 (revised in 2002). Informed consent was obtained from each patient before the study. 

To be included in the study, patients had to be Caucasian and > 18 years of age and to have been diagnosed as suffering from stage III periodontitis according to the clinical and radiographic criteria by the 2018 classification of periodontal diseases [7]. In addition, they were required to have at least 20 natural teeth and demonstrate a minimum of 12 teeth with PD and interdental clinical attachment level (CAL) ≥ 5 mm, and radiographic evidence of alveolar bone loss extending at least to the middle part of the root. 

Patients were divided into two groups according to the risk of periodontitis progression. Group 1 included patients classified by a moderate rate of progression (stage III, grade B periodontitis) based on CAL loss less than 2 mm over the past 5 years or the bone/age ratio (BL/A) from full-mouth radiographs between 0.25 and 1. Group 2 included patients with a rapid rate of progression (stage III, grade C periodontitis) in which CAL loss was at least 2 mm or BL/A was higher than 1. Exclusion criteria for both groups were current or past smoking, pregnancy, lactation, allergy, asthma, periodontal treatment or/and antibiotic therapies in the previous 6 months, and any systemic disease that could influence the course of periodontal disease. Subjects using anti-inflammatory and immunosuppressive medications, or any other medications known to affect periodontal status, were also excluded.

### 2.2. Clinical Examination and Periodontal Treatment

All enrolled subjects underwent periodontal examination by an experienced periodontist. After calibration, a 95.9% and 97.2% concordance within 1 mm for measurements of PD and CAL, respectively, between the first and the second recording with an interval of 24 h was reached. Clinical measurements were taken at six sites per tooth of every tooth present, except third molars, with a standardized periodontal probe (PCP UNC15, Hu-Friedy, Chicago, IL, USA) and included presence of plaque (PI), bleeding on probing (BoP), gingival index (GI), PD, and CAL. Full-mouth percentage of sites with PI (full-mouth plaque score, FMPS) and BoP (full-mouth bleeding score, FMBS) was also recorded. Full-mouth periapical radiographs were taken with the long cone paralleling using Rinn holders. 

After baseline examination, both group subjects underwent a session of supragingival scaling and polishing and received instructions in proper self-performed plaque control measures, including tailored instructions in electric toothbrush technique and interproximal cleaning with dental floss and interdental brushes. One week later, patients underwent quadrant-wise full-mouth subgingival scaling and root planing (SRP) in four sessions using hand instruments (Gracey curettes, Hu-Friedy, Chicago, IL, USA) and ultrasonic scalers (Cavitron Select, Dentsply, York, PA, USA). Subgingival instrumentation was performed under local anesthesia without a time limit until the root surface felt smooth and clean to an explorer tip. The entire non-surgical periodontal treatment was completed in 28 days without the administration of local and/or systemic antimicrobials. Supportive therapy, including professional plaque control and reinstruction of oral hygiene, was performed on a 2-week interval during the first 6 weeks postoperatively and every 2 months up to the 6-month re-evaluation. 

### 2.3. GCF Sampling and Multiplex Bead Immunoassay Analysis

GCF samples were collected as previously described [14]. Briefly, they were withdrawn from two inflamed sites with moderate (4 to 5 mm) and deep PD (≥6 mm) on the mesial aspect of anterior periodontally involved teeth in both periodontitis groups. Sites showing the greatest PD along with radiographic bone loss were selected for sampling. The same sites were sampled at baseline and 6 months post-surgery. Sites to be sampled were isolated with cotton after removing the supragingival plaque and the crevicular area was gently dried with an air syringe. GCF samples were collected by inserting two paper strips (PerioPaper Strips, Oraflow Inc., Plainview, NY, USA) into the pocket for 30 s. Strips contaminated with blood and saliva were discarded. The amount of collected GCF was measured using an electronic device Periotron 8000 (Oraflow Inc., Plainview, NY, USA), which was calibrated as reported in detail elsewhere [30]. All strips with GCF were placed separately into coded sealed Eppendorf tubes containing 100 µL of sterile phosphate-buffered saline (PBS) and stored at −80 °C according to the method previously published [22].

A single investigator blinded to the clinical data performed the GCF cytokine quantification. IL-1β, TNF-α, IL-9, and VEGF amounts were detected simultaneously in biological samples by means of a high-sensibility Bio-Plex Suspension Array System (Bio-Rad Laboratories S.r.l., Segrate, Milan, Italy) according to the manufacturer’s instructions. Briefly, opportune anticytokine antibody-conjugated beads were loaded into individual wells of a 96-well plate. After washing, standards and GCF undiluted samples were added into respective wells and incubated for 30 min. After plates were washed, biotin-conjugated detection antibody was added. After another 30 min of incubation and consequent washing, streptavidin-conjugated PE was added for 10 min. After an additional wash, the complex was solubilized by adding the Bio-Plex assay buffer to each well. Then, plates were analyzed with the Bio-Plex Suspension Array System. Total amounts (pg/site) of each cytokine were determined. 

### 2.4. Statistical Analysis

Statistical analyses were performed using software (SPSS Statistics for Mac, v. 24.0, IBM, Chicago, IL, USA). The primary outcome measure of the study was variation in IL-1β levels after treatment. Secondary outcomes included changes of GCF levels of IL-9, TNF-α VEGF as well as changes in clinical parameters. Based on a pilot study including five subjects in each group, the sample size was estimated at 15 subjects in each group to achieve 80% power to detect a minimum expected difference of 20 pg in IL-1β levels with at 0.05 two-sided alpha error. For compensation of possible dropouts, a total of 40 individuals were recruited. 

Clinical and biochemical data were first examined for normality with the Shapiro–Wilk test and if they did not achieve normality, analyses were performed using non-parametric methods. The intragroup changes in clinical parameters were analyzed by means of the paired Student *t*-test (PD, CAL) or Wilcoxon signed rank test (FMPS, FMBS, GCF volume). In addition, the Wilcoxon signed rank test was employed to detect significant cytokine difference within periodontitis groups prior to and after non-surgical periodontal treatment. Comparisons in clinical and GCF biochemical parameters between periodontitis groups at baseline and 6 months post-therapy were performed by means of unpaired Student *t*-test or Mann–Whitney U-test, as appropriate. A chi-square analysis was used to compare sex ratio between groups. The correlations between BL/A and GCF cytokine levels were analyzed with the Spearman correlation coefficient. To estimate the diagnostic potential of the quantified individual cytokines to discriminate between groups, a receiver-operating characteristic (ROC) analysis was carried out. The ROC curve was also used to identify the best cut-off value that maximizes the difference between true positive subjects and false positives ones [31]. Finally, diagnostic models for Grade C periodontitis were constructed according to the total number of cytokines based on a forward stepwise logistic regression analysis and were evaluated by ROC curve analysis. The significance level was set at 5%.

## 3. Results

Sixty-seven patients suffering from Stage III periodontitis were consecutively selected for enrolment. Eighteen subjects did not meet the inclusion criteria and other nine did not attend the baseline examination. Twenty subjects with grade B periodontitis (7 males and 13 females, mean age 53.2 ± 7.5 years) and twenty patients with grade C periodontitis (9 males and 11 females, mean age 37.3 ± 5.9 years) were joined and completed the trial. No previous patients’ records were available; thus, the rate of disease progression was based on BL/A ratio (0.9 ± 0.1 vs. 1.4 ± 0.2) 

Demographic characteristics were comparable between the groups excluding age that was significantly higher in the grade C group (*p* < 0.001).

### 3.1. Clinical Findings

Clinical parameters at baseline and 6 months post-therapy are summarized in Table 1. Statistically significant improvement in the overall mean examined clinical parameters was observed in both clinical groups at 6 months post-therapy (*p* < 0.001). No statistically significant differences were observed between groups.

Similarly, between-group analysis showed statistically significant reduction in clinical parameter scores during the experimental period in both moderate and deep pocket sites selected for GCF sampling (*p* < 0.01) (Table 2). PD reduction and CAL gain were significantly greater at deep pockets at 6 months of examination (*p* < 0.001). Non-surgical periodontal treatment was also associated with a significant decrease in mean GCF volumes that was more evident in deep sites (*p* < 0.001). The mean changes in clinical parameters did not significantly differ between periodontitis groups.

### 3.2. Biochemical Findings

SRP was effective in reducing IL-1β total amounts (Figure 1) in both moderate and deep sites (*p* < 0.001 for Grade C and *p* < 0.05 for Grade B, respectively) and TNF-α levels (Figure 2) in deep sites (*p* < 0.05) of both periodontitis groups.

GCF total amounts of both biomarkers were found significantly higher at baseline in moderate sites of grade C periodontitis subjects when compared with the sites from grade B periodontitis patients (*p* < 0.01). At 6 months after non-surgical treatment, they were still higher, but only IL-1β levels reached statistical significance (*p* < 0.05) (Figure 1A and Figure 2A). On the contrary, no statistically significant differences were observed among deep pockets from the experimental groups over the study period (Figure 1B and Figure 2B).

The GCF amounts of VEGF (Figure 3) were similar at baseline between the periodontitis groups and decreased in deep sites of both groups (*p* < 0.05) at 6 months post-therapy but the reduction was significantly higher in grade B periodontitis patients (*p* < 0.01).

Concerning the IL-9 levels of moderate and deep sites, no significant changes were detected after treatment within each periodontitis group (Figure 4). Higher amounts of IL-9 were observed at baseline and 6 months after SRP (*p* < 0.01) in moderate sites of grade B periodontitis compared with grade C periodontitis patients (Figure 4A), while no differences were found in deep pocket sites (Figure 4B).

### 3.3. Correlation and ROC Analysis

IL-1**β** levels showed a significant and positive correlation with BL/A at baseline (r = 0563, *p* < 0.001) and also after a 6-month evaluation (r = 0.361, *p* < 0.05) in moderate pockets, and VEGF with BL/A only at 6 months in deep sites (r = 0.366, *p* < 0.05). A significant negative correlation was found between BL/A and GCF amount of IL-9 in moderate sites at baseline (r = −0.378, *p* < 0.05) and 6 months after SRP (r = −0.367, *p* < 0.05). 

Because association analyses are not sufficient to identify possible diagnostic and prognostic markers, we determined cut-off values for these cytokines using ROC analysis, which showed low values for the areas under the curve (AUC) for IL-9 (AUC < 0.50). VEGF showed AUC of 0.76 (AUC = 0.76, 95% IC: 0.59–0.93, *p* < 0.01) with moderate sensitivity (0.65) and low specificity (0.41). Finally, the AUC was 0.88 (95% CI: 0.772–0.998, *p* < 0.001) for IL-1β, which suggests that CGF amounts of IL-1β ≥ 115.26 pg in moderate sites identify the presence of highly progressive periodontitis, with a sensitivity of 0.94 and a specificity of 0.70 (Figure 5). 

Diagnostic model of Grade C periodontitis was constructed combining IL-1β and IL-9 GCF levels and showed an accuracy of 0.92 with high sensitivity (0.94) and specificity (0.89). The AUC value was 0.94. The mathematical formula was: $$\mathrm{Grade}\;C\;periodontitis\;=\frac{1}{1+e^{-(1.473+0.22IL-1-6.35IL-9)}}$$

## 4. Discussion

To the best of knowledge of the authors, this is the first study evaluating the GCF cytokine levels of subjects suffering from stage III periodontitis with moderate and rapid rates of progression before and after non-surgical periodontal treatment. The GCF amounts of IL-1β, IL-9, TNF-α, and VEGF were measured using a highly specific and sensitive multiplex bead immunoassay that allows for analyzing multiple biomarkers in the same sample even in a small amount of liquid. Using this approach, it is possible to study the role of inflammatory mediators in periodontitis that are thought to function in complex networks [22]. Most of the previous studies investigated cytokine profile using commercial immunoenzymatic assays [32]. These methods usually require higher volumes of GCF and therefore it needs to resort to serial withdrawals, which could be source of mistakes. 

Several inflammatory cytokines are released by the host in response to subgingival microbiota, which in turn lead to clinical and radiographical signs of periodontal disease progression [10,32]. Among these cytokines, IL-1β and TNF-α, when released in high concentration, can stimulate the production of other inflammatory mediators involved in extracellular matrix connective tissue destruction and osteoclastic-mediated bone loss, as well as upregulate the production of angiogenic mediators [24,25]. In the regulation of angiogenesis and vascular permeability, VEGF is a crucial mediator. In fact, the production of VEGF is increased in sites with higher severity of periodontal disease [21]. Their concomitant presence in GCF and the juxtaposed diseased periodontal tissue denotes that GCF reflects the periodontal inflammatory status [33]. Since changes in GCF volume influence the GCF component concentrations, the present study was based on total amounts. Previous studies showed that considering biomarkers in terms of total amounts of GCF per sampling time in the GCF, rather than in terms of concentration is more reliable [34,35].

In the current study, patients underwent conventional staged quadrant-wise debridement and strict plaque control (FMPS < 15%) was maintained through the study period. The overall and site-specific improvements in clinical inflammatory and periodontal disease parameters were comparable in both treatment groups and in line with those reported in the literature [36]. This points out the importance of adequate home oral hygiene performance and tailored professional oral hygiene sessions after active treatment.

Consistent with our hypothesis, the trend of GCF inflammatory-related mediators after non-surgical periodontal treatment differed by severity of periodontal damage and risk of disease progression. VEGF levels were found to be reduced compared to baseline values in deep pocket sites of both periodontitis groups, but were decreased more in slowly progressive periodontitis. These results partially agree with findings of a companion paper that demonstrated a reduction in VEGF levels in both moderate and deep pocket sites in aggressive periodontitis patients [22]. Since disease progression implies active expansion of the vasculature network, these findings may suggest a potential lower expression of VEGF in these individuals less susceptible to developing further periodontal breakdown [37]. However, the role of VEGF in periodontal inflammation remains unclear [38,39]. Although higher expression has been shown during healing period of periodontal disease [39,40], many studies reported elevated VEGF levels in periodontitis sites compared with gingivitis and healthy gingiva [19,20,21]. 

In this high activity scenario, the reduction in IL-1β in both moderate and deep sites and TNF-α in deep sites of both periodontitis groups suggests a role for these cytokines in the disease process [24]. Previous studies showed an association between PD and CAL and GCF IL-1β levels in patients with severe periodontitis [3,41] and reported a significant reduction in this cytokine after SRP [12,13,14,15,42]. Interestingly, in this study, it was observed that IL-1β levels were still high in moderate PD sites of highly progressive patients over the 6-month follow-up, although both periodontitis groups had similar GI and BoP scores. It should be noted that these parameters characterize the degree of clinical gingival inflammation and may not detect the presence of subgingival inflammation in the adjacent periodontal tissue [43]. Thus, elevated amounts of IL-1β may indicate a periodontal microenvironment more prone to disease progression. This cytokine stimulates the action and coordinates the course of the immune response [26]. Martinez-Villa et al. reported a positive correlation between GCF levels of IL-1β and percentage of plasma cells in shallow sites after non-surgical periodontal treatment [44].

Evidence for using TNF-α to assess therapeutic outcomes is instead contradictory. In the present study, TNF-α levels were significantly decreased in GCF of deep sites in both periodontitis groups, but they did not change in moderate sites. In contrast, other studies reported that TNF-α levels remained stable or increased after mechanical periodontal therapy but did not stratify data on PD severity [14,45]. According to a recent systematic review, TNF-α seems to be a reliable biomarker for diagnosis of periodontal disease but not to monitor the periodontal healing [46]. Bastos et al. showed high levels of this mediator in both healthy and diseased sites from an aggressive form of periodontitis, indicating that TNF-α can be expressed in sites with different clinical statuses [47].

Although the role of IL-9 in the periodontal diseases pathogenesis is unclear, some studies have shed light on its role in other chronic inflammatory diseases. Th9 lymphocytes and higher levels of IL-9 have been detected in patients treated for rheumatoid arthritis and in inactive periapical lesions, suggesting a potential role in tissue healing by modulating Th1-mediated osteoclastic activity [28,29,48]. A significant increase in IL-9 concentration was also observed at moderate pocket sites 3 months after the completion of non-surgical periodontal therapy in patients with aggressive periodontitis [20]. Recently, Díaz-Zúñiga et al. detected an overexpression of IL-9 in GCF and gingival samples obtained from subjects with gingivitis compared with healthy individuals, but no differences between patients with gingivitis and periodontitis were reported [49]. Furthermore, no significant correlation was found between GCF IL-9 and CAL in periodontitis patients [49]. IL-9 has been described as pleiotropic cytokine, whose pro- or anti-inflammatory activity may differ depending on the overall cytokine milieu [29,50]. 

We found significantly higher amounts of IL-9 at baseline and 6 months after SRP in moderate sites of grade B periodontitis compared to grade C periodontitis patients, while no differences in deep pocket sites were found. It is possible that Th9 lymphocytes overexpression could translate clinically into decreased inflammation and bone resorption [49]. Taken together, these findings may suggest that the inflammatory fingerprint characteristic of individual patients is more likely expressed in moderate pocket sites and also tends to be expressed following the control of etiological factors. In deep pockets, that in the current study were more than 7 mm deep, it should be hypothesized that the environmental conditions have a more relevant role in up-regulating cytokine expression against local dysbiotic bacteria prior to and after the non-surgical treatment. 

The diagnostic accuracy of the different cytokines in terms of increased amount was evaluated by the ROC curve analysis to discriminate between rates of disease progression. We found that IL-1β was the most important biomarker associated with the disease. Quantification of IL-1β in moderate sites using a sensitive detection technique showed a good performance with an AUC > 0.8. A cut-off value of 125.26 pg/site correctly classified 94% of quickly progressing and 70% of slowly progressing patients. A recent study using diagnostic models to classify chronic periodontitis from healthy controls based on the number of amounts obtained similar percentages of sensitivity and specificity using GCF samples [51]. Interestingly, the ability of IL-1β to discriminate between periodontitis groups decreased when considering GCF levels in deep sites. These findings would seem to confirm that moderate sites mirror the individual inflammatory signature of periodontitis patients. Despite the higher post-therapy expression of IL-1β in moderate sites of highly progressive patients, the ROC analysis showed small discrimination accuracy in stage III periodontitis. It should be considered that mean value could not adequately characterize the site-specificity of periodontal disease.

The combination of IL-1β and IL-9 GCF levels slightly increased the AUC value to 0.94 due to higher specificity of the diagnostic model as compared to the AUC value of IL-1β alone, while the sensitivity remained nearly unchanged. 

The present research has some limitations. Although we are in a scenario of small data, the sample size used was large enough to detect statistically significant differences between groups. In addition, data were short-term. Furthermore, we analyzed only a limited panel of biomarkers and did not relate it to the GCF microbiota. Dysbiosis of the oral microbiome can induce the immune reaction of the host, resulting in the release of various inflammatory mediators. Based on recent reports there is a strict relationship between microbial composition and cytokines pattern [52,53], thus the combined profile of GCF biomarkers and periodontal pathogens might provide a more sensitive approach to detect disease progression [54]. Finally, the study population consisted of only Caucasian and non-smoking periodontitis patients. This allows the confounding effects of ethnic background and smoking on inflammatory markers to be avoided [55,56,57], but it may limit the generalizability of the present findings to other populations. 

## 5. Conclusions

The present study provided evidence for the first time on distinct patterns of cytokine expression in moderate sites of grade C compared to grade B periodontitis patients, while no differences were detected in deep sites. Despite the similar clinical improvement in response to non-surgical periodontal treatment, moderate pockets of grade C periodontitis patients still presented higher IL-1β and lower IL-9 GCF levels. It is currently accepted that locally produced pro-inflammatory cytokines contribute to disease progression, indicating them as putative periodontal disease biomarkers. 

Since clinical criteria are inappropriate to measure the degree of susceptibility to progression, IL-1β may be a useful biomarker to improve the early assessment of periodontitis patients at high risk of future breakdown. A mathematical model was also constructed based on the subjects’ GCF levels of both IL-1β and IL-9 GCF. It could help clinicians to select the appropriate approach for periodontitis patients individually to prevent periodontitis progression.

Confirmatory studies with a larger number of patients and longitudinal evaluations with longer follow-up are required to prove the reliability of these biomarkers in discriminating the rate of periodontitis progression.

Due to the small volume of GCF, it is necessary to use a combined approach (involving the combination of Periotron with the high-sensitivity Bio-Plex Suspension Array System) to analyze different biomarkers together in the same sample even if in very low quantity. Although molecular analysis of GCF is promising, it is time consuming and laboratory based and it depends also on both sites selected for sampling and the sampling methods employed. Still, this limits the applicability of this approach in the daily clinical practice [33].

## Figures and Tables

**Figure 1 biomedicines-08-00515-f001:**
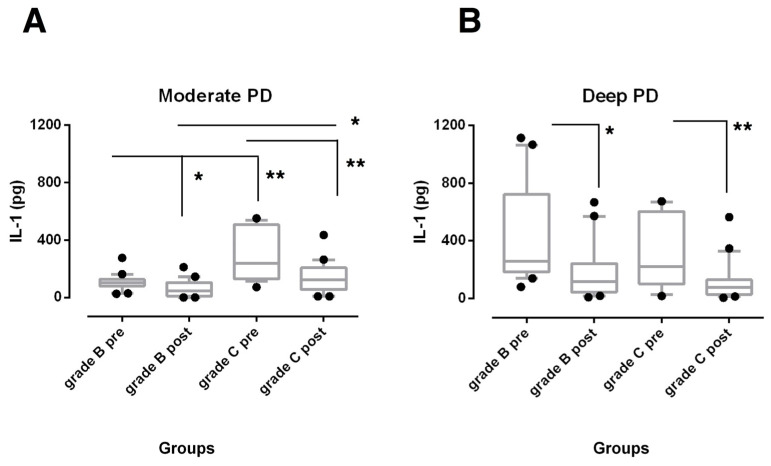
Box-and-whisker plots showing the total amount of interleukin (IL)-1β in gingival crevicular fluid of moderate (**A**) and deep pocket sites (**B**) in subjects with grade B and grade C stage III periodontitis before (pre) and after non-surgical periodontal therapy (post). Horizontal line represents the median with the box representing the 25th and 75th percentiles, and the whiskers denote the 5th and 95th percentiles. * *p* < 0.05, ** *p* < 0.01.

**Figure 2 biomedicines-08-00515-f002:**
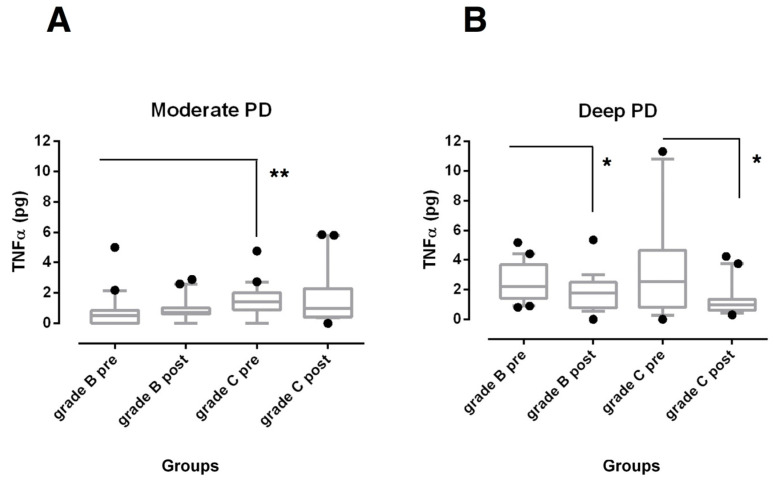
Box-and-whisker plots showing the total amount of tumor necrosis factor (TNF)-α in gingival crevicular fluid of moderate (**A**) and deep pocket sites (**B**) in subjects with grade B and grade C stage III periodontitis before (pre) and after non-surgical periodontal therapy (post). Horizontal line represents the median with the box representing the 25th and 75th percentiles, and the whiskers denote the 5th and 95th percentiles. * *p* < 0.05, ** *p* < 0.01.

**Figure 3 biomedicines-08-00515-f003:**
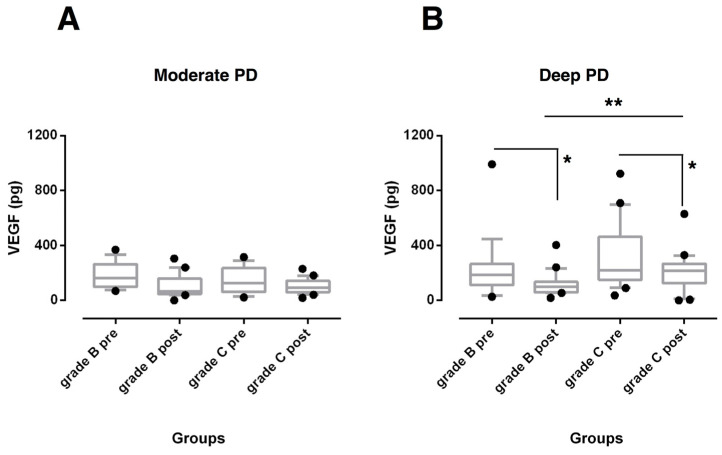
Box-and-whisker plots showing the total amount of vascular endothelial growth factor (VEGF) in gingival crevicular fluid of moderate (**A**) and deep pocket sites (**B**) in subjects with grade B and grade C stage III periodontitis before (pre) and after non-surgical periodontal therapy (post). Horizontal line represents the median with the box representing the 25th and 75th percentiles, and the whiskers denote the 5th and 95th percentiles. * *p* < 0.05, ** *p* < 0.01.

**Figure 4 biomedicines-08-00515-f004:**
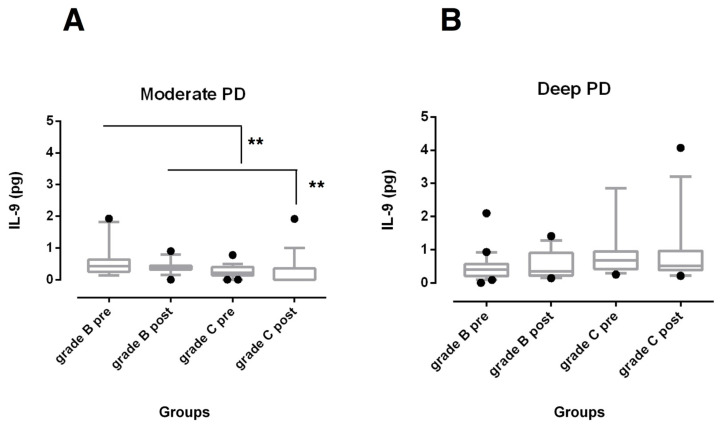
Box-and-whisker plots showing the total amount of IL-9 in gingival crevicular fluid of moderate (**A**) and deep pocket sites (**B**) in subjects with grade B and grade C stage III periodontitis before (pre) and after non-surgical periodontal therapy (post). Horizontal line represents the median with the box representing the 25th and 75th percentiles, and the whiskers denote the 5th and 95th percentiles. ** *p* < 0.01.

**Figure 5 biomedicines-08-00515-f005:**
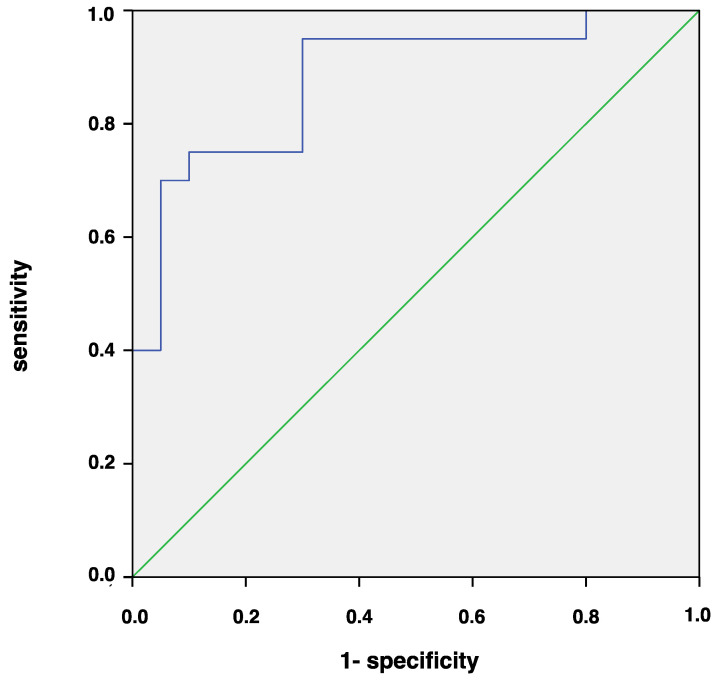
Receiver operating characteristic (ROC) curve for IL-1β amount in moderate pocket sites comparing stage III periodontitis patients with rapid (Grade C) versus moderate (Grade B) rates of progression.

**Table 1 biomedicines-08-00515-t001:** Changes in clinical variables (mean ± SD) over the 6-month experimental period (full-mouth data).

Variables	Group	Baseline	6 Months	Δ_0–6 months_
FMPS (%)	Stage B	71.7 ± 15.9	14.6 ± 4.9 ***	57.1 ± 17.7
	Stage C	74.5 ± 25.6	13.3 ± 4.6 ***	61.2 ± 24.8
Difference between groups		NS	NS	
FMBS (%)	Stage B	56.0 ± 20.7	13.6 ± 4.8 ***	42.4 ± 19.6
	Stage C	63.8 ± 23.0	16.1 ± 4.4 ***	47.7 ± 22.7
Difference between groups		NS	NS	
PD (mm)	Stage B	4.6 ± 0.7	3.4 ± 0.4 ***	1.2 ± 0.5
	Stage C	4.9 ± 0.5	3.6 ± 0.6 ***	1.3 ± 0.4
Difference between groups		NS	NS	
CAL (mm)	Stage B	5.3 ± 1.0	4.4 ± 0.8 ***	0.9 ± 0.4
	Stage C	5.6 ± 0.9	4.5 ± 0.7 ***	1.2 ± 0.6
Difference between groups		NS	NS	

SD: standard deviation; FMPS: full-mouth plaque score; FMBS: full-mouth bleeding score; PD: probing depth; CAL: clinical attachment level; NS: difference between groups is not statistically significant (*p* > 0.05, Mann–Whitney U-test or unpaired *t*-test); *** *p ≤* 0.001, *p* values represent changes from baseline (Wilcoxon test or paired *t*-test).

**Table 2 biomedicines-08-00515-t002:** Changes in clinical variables (mean ± SD) over the 6-month experimental period (gingival crevicular fluid (GCF) sampling sites).

Variables	Group	Moderate Pocket Sites (4–5 mm)		Deep Pocket Sites (≥6 mm)	
		Baseline	6 Months	Baseline	6 Months
PI	Stage B	85.0 ± 23.5	10.0 ± 20.5 ***	92.5 ± 18.3	25.0 ± 30.3 ***
	Stage C	77.5 ± 25.5 ^¶^	15.0 ± 28.6 ***	95.0 ± 15.4	20.0 ± 29.9 ***
Difference between groups		NS	NS	NS	NS
GI	Stage B	2.0 ± 0.5 ^¶^	0.6 ± 0.5 ***	2.6 ± 0.5	0.9 ± 0.8 ***
	Stage C	1.8 ± 0.4 ^¶^	0.4 ± 0.6 ***	2.4 ± 0.4	0.8 ± 0.7 ***
Difference between groups		NS	NS	NS	NS
PD (mm)	Stage B	4.7 ± 0.5	3.0 ± 1.0 ***	8.2 ± 1.4	4.5 ± 0.9 ***^,§^
	Stage C	4.8 ± 0.4	3.2 ± 0.6 ***	8.9 ± 1.7	4.9 ± 1.0 ***^,§^
Difference between groups		NS	NS	NS	NS
CAL (mm)	Stage B	5.8 ± 1.4	4.5 ± 1.8 **	9.3 ± 2.1	6.4 ± 2.1 **^,§^
	Stage C	5.3 ± 0.7	4.1 ± 1.1 ***	9.6 ± 1.9	6.6 ± 1.4 ***^,§^
Difference between groups		NS	NS	NS	NS
GCF (µL)	Stage B	0.56 ± 0.36	0.22 ± 0.15 **	1.09 ± 0.25 ^§^	0.45 ± 0.21 ***^,§^
	Stage C	0.59 ± 0.39	0.27 ± 0.20 *	1.13 ± 0.33 ^§^	0.54 ± 0.31 **^,¶^
Difference between groups		NS	NS	NS	NS

SD: standard deviation; GCF: gingival crevicular fluid; PI: presence of plaque; GI: gingival index; PD: probing depth; CAL: clinical attachment level; NS: difference between groups is not statistically significant (*p* > 0.05); * *p* < 0.05 versus baseline; ** *p* < 0.01 versus baseline. *** *p* < 0.001 versus baseline; ^¶^
*p* < 0.01 versus moderate pocket sites; ^§^
*p* < 0.001 versus moderate pocket sites.

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
