# Peer review of "Gingival Crevicular Fluid Cytokines in Moderate and Deep Sites of Stage III Periodontitis Patients in Different Rates of Clinical Progression"

_biomedicines, 2020, doi:10.3390/biomedicines8110515_

Round 1
Reviewer 1 Report
Abstract:
I dont understand what you mean by severe periodontitis? a clinical description based on a classification or due to its consequences on the esthetic, function, ...?
There is no need to mention periotron
Introduction:
l61: its role
l78: what would be the clinical consequences of their early identification? what will change in their management in comparison with other patients?
l83: what is the originality of the study? several trials have been conducted to assess the cytokines levels before and after non surgical treatment.
Materials and methods:
l93: why only caucasian?
Results section is well written
Can you present the ROC curve with the points corresponding to the patients?
Discussion:
l372: GCF
Discussion is well written, however, I suggest to highlight the limitations of the use of GCF cytokines as biomarkers. Over the last 20 years, research has been conducted to determine their possible clinical use but, till today, none were considered clinically relevant due to the complexity of the analysis and the influence of site. Please discuss it.
Author Response
Turin, 12th November 2020
Dear Editor,
Thank you for the comments regarding our manuscript (biomedicines 938061) entitled “Gingival crevicular fluid cytokines in moderate and deep sites of stage III periodontitis patients in different rate of clinical progression.” by F. Romano et al.
The Reviewers comments spurred us to further improve the overall quality of our message.
Yours faithfully
Giovanni N Berta
Authors’ responses to Reviewers:
Reviewer #1:
Comment 1 to the Authors: Abstract. I don’t understand what you mean by severe periodontitis? a clinical description based on a classification or due to its consequences on the esthetic, function, ...?
Authors’ response/action: We referred to the current classification of periodontitis (Tonetti et al. 2018) that described stage III periodontitis as severe periodontitis with potential for additional tooth loss. We have replaced “severe periodontitis” with “stage III periodontitis” for clarity.
Comment 2 to the Authors: Abstract. There is no need to mention periotron
Authors’ response/action: We have deleted from the abstract the use of the periotron as suggested.
Comment 3 to the Authors: Introduction. l61: its role
Authors’ response/action: We apologize for the mistake. We have corrected it in the text.
Comment 4 to the Authors: Introduction.l78: what would be the clinical consequences of their early identification? what will change in their management in comparison with other patients?
Authors’ response/action:
Conventionally, disease progression is measured by recording clinical parameters. Indeed, at least 2 mm of CAL loss are required to consider a periodontal site as rapidly progressing according to the current classification of periodontal diseases (Tonetti et al. 2018). This method of monitoring disease breakdown enables the clinician to assess the history of the disease rather than to perform a real-time appraisal of disease activity. In this context, analysis of CGF biomarkers may contribute to the detection of real-time changes within the periodontal tissues. Identification of tooth sites that, in spite of the resolution of clinical signs of inflammation, still harbour a pro-inflammatory environment may have relevant implications into the clinical practice improving the clinical monitoring and thus the long-term outcomes of periodontal treatment in high-risk patients. We have discussed this point between lines 77 and 82.
Comment 5 to the Authors: Introduction. l83: what is the originality of the study? several trials have been conducted to assess the cytokines levels before and after non surgical treatment.
Authors’ response/action: In introduction section we have better defined the originality of our study (lines 74-77)
Comment 6 to the Authors: Mat and Met. l93: why only caucasian?
Authors’ response/action: Thanks for this remark. Different racial and ethnic groups within a given population often show marked differences in inflammatory response and periodontal diseases outcome (Albandar 2002). For instance Blacks have been shown to exhibit higher levels of CRP and pro-inflammatory cytokines than Whites. Ethnic specific differences in inflammation may have diverse determinants. Behavioral factors and socioeconomic status have been associated to increased levels of CRP, IL-6 and TNF-alpha (Koster et al. 2006, Stowe et al. 2009). Genetics may also exert a direct impact on inflammatory processes. Gene polymorphism may act to modulate production of pro-inflammatory mediators (i.e. IL.1 and TNF-alpha). Eickholz et al. (2013) reported less pronounced CRP and elastase reduction 12 weeks after scaling and root planing in Caucasian periodontitis patients as compared to those with African origin. We have discussed this aspect between lines 397 and 400.
Comment 7 to the Authors: Results. Can you present the ROC curve with the points corresponding to the patients?
Authors’ response/action: We have added the ROC curve as new figure (Figure 5).
Comment 8 to the Authors: Discussion. l372: GCF.
Authors’ response/action: We apologize for the mistake. We have corrected it in the text.
Comment 9 to the Authors: Discussion is well written, however, I suggest to highlight the limitations of the use of GCF cytokines as biomarkers. Over the last 20 years, research has been conducted to determine their possible clinical use but, till today, none were considered clinically relevant due to the complexity of the analysis and the influence of site. Please discuss it.
Authors’ response/action: We are grateful with the Reviewer to arise this point. We have discussed this aspect between lines 419 and 422 in the text: “Although molecular analysis of GCF is promising, it is time consuming and laboratory based and it depends also on both sites selected for sampling and sampling methods employed. Still, this limits the applicability of this approach in the daily clinical practice [58]”.
Reviewer 2 Report
The objective of this study is to search for a possible cytokine biosiganture predictive of the response to periodontal treatment and the progression of periodontal disease.
The methodology used is fairly standard with two groups of 20 patients, excluding smokers, which should be emphasized and given to the credit of this work since it makes it more difficult to recruit patients but it reduces the confounding factors linked to smoking.
Comments :
- The authors exclude the influence of bacterial biofilm with 2 references from 1965 and 2004. They should integrate recent data concerning the analysis of the periodontal microbiota with more powerful techniques such as 16s RNA analysis, shotgun etc ... and justify why they did not take microbial samples from the same sites to study the association between certain bacterial species and cytokine expression in order to have a more exhaustive perspective
- The authors justify their choice of cytokines but we do not know why they did not extend to other cytokines with the technique used which would have made it possible to investigate cytokines such as IL-6, IL-10, MMPs, and other cytokines. In addition, it would have been relevant to take samples of gingival fluid from healthy sites in all patients in order to define a kind of basic profile of cytokine expression and compare it to that of moderate and deep sites in the same patients.
- how were the authors able to determine the progression of periodontal disease (CAL and bone loss) during the last 5 years?
Author Response
Turin, 12th November 2020
Dear Editor,
Thank you for the comments regarding our manuscript (biomedicines 938061) entitled “Gingival crevicular fluid cytokines in moderate and deep sites of stage III periodontitis patients in different rate of clinical progression.” by F. Romano et al.
The Reviewers comments spurred us to further improve the overall quality of our message.
Yours faithfully
Giovanni N Berta
Reviewer #2:
Comment 1 to the Authors: The methodology used is fairly standard with two groups of 20 patients, excluding smokers, which should be emphasized and given to the credit of this work since it makes it more difficult to recruit patients but it reduces the confounding factors linked to smoking.
Authors’ response/action: Thanks for this remark. It has been widely demonstrated that smoking is strongly involved in the onset and progression of periodontitis and that smokers exhibit altered amount of GCF volume and composition (Mokeem et al., 2014; Kubota et al., 2011). For all these reasons we excluded smokers from this study. However, as reported between lines 397 and 400 in the discussion, this may have affected the generalizability of the present results.
Comment 2 to the Authors: The authors exclude the influence of bacterial biofilm with 2 references from 1965 and 2004. They should integrate recent data concerning the analysis of the periodontal microbiota with more powerful techniques such as 16s RNA analysis, shotgun etc ... and justify why they did not take microbial samples from the same sites to study the association between certain bacterial species and cytokine expression in order to have a more exhaustive perspective.
Authors’ response/action: Thanks to the Reviewer for raising this issue. We have added new references in the introduction and we have considered the lack of microbiological data as limitation of the present study. In particular, we added a new phrase in the discussion section between lines 392 and 397: “Furthermore, we analysed only a limited panel of biomarkers and did not relate it to the GCF microbiota. Dysbiosis of the oral microbiome can induce the immune reaction of the host, resulting in the release of various inflammatory mediators. Based on recent reports there is a strict relationship between microbial composition and cytokines pattern [52,53], thus the combined profile of GCF biomarkers and periodontal pathogens might provide a more sensitive approach to detect disease progression [54]”.
Comment 3 to the Authors: The authors justify their choice of cytokines but we do not know why they did not extend to other cytokines with the technique used which would have made it possible to investigate cytokines such as IL-6, IL-10, MMPs, and other cytokines. In addition, it would have been relevant to take samples of gingival fluid from healthy sites in all patients in order to define a kind of basic profile of cytokine expression and compare it to that of moderate and deep sites in the same patients.
Authors’ response/action: This is a very interesting question. Different cytokines have been proposed in the literature to detect alterations in tissue metabolism, inflammatory-cell recruitment and connective tissue remodeling in periodontally involved sites (Barros et al. 2016). The goal of the present study was to analyse the pro-inflammatory profile of stage III periodontits patients with grade B and C in relation to the severity of initial peridontal damage before and after non-surgical periodontal treatment. Thus, we selected IL-1 and TNF-alpha and VEGF. These cytokines play a pivotal role in chronic inflammation and stimulate the production of secondary mediators responsible for the degradation of connective tissue and osteoclastic bone resorption. We were interested also to assess whether IL-9 could be play a role in periodontal healing. Previous studies reported that it increased in inactive periapical lesions and suggested a protective role by modulating Th1-mediated osteoclastic activity. We collected GCF from moderate and deep diseased sites but not from healthy sites in periodontitis patients. Sites showing the greatest PD along with radiographic bone loss were selected for sampling. Most of the studies in the literature assessed cytokine changes after treatment of chronic and aggressive periodontitis without differentiating among different PDs. In a few studies reporting levels of IL-1 and TNF-alpha in healthy sites, they were higher than those detected in diseased sites and did not vary significantly after periodontal treatment (Toker et al. 2008; Thunell et al. 2010; Rescala et al. 2010).
In addiction there is a “technical” motivation: not all cytokines are detectable in tha same bead immunoassay; for example MMPs analysys required different multiwell-plate (due to the different reagents utilized).
Comment 4 to the Authors: how were the authors able to determine the progression of periodontal disease (CAL and bone loss) during the last 5 years?
Authors’ response/action: According to the diagnostic criteria of the current classification (Tonetti et al. 2018) we assessed at the initial examination the rate of disease progression based on the ratio between severity of radiographic bone loss and age due to the absence of direct evidence. In fact, we did not have clinical recordings/ radiographs of all the included patients allowing comparison of CAL or marginal bone loss over time. We have elucidated this point in the Results section (lines 196-197).
Reviewer 3 Report
Authours reported that a mathematical model was constructed to identify Grade C periodontitispatients based on the subjects’ GCF levels of IL-1β and IL-9 GCF, which achieved an area under the ROC curve of 0.94.
I'm not sure what the novelty in this study is.
What is the novelty in this study?
It's better to be clear.
Author Response
Turin, 12th November 2020
Dear Editor,
Thank you for the comments regarding our manuscript (biomedicines 938061) entitled “Gingival crevicular fluid cytokines in moderate and deep sites of stage III periodontitis patients in different rate of clinical progression.” by F. Romano et al.
The Reviewers comments spurred us to further improve the overall quality of our message.
Yours faithfully
Giovanni N Berta
Reviewer #3:
Comment 1 to the Authors: I'm not sure what the novelty in this study is. What is the novelty in this study?
It's better to be clear
Authors’ response/action: In introduction section we have better defined the originality of our study (lines 75-82); see answer #5 Reviewer 1
Round 2
Reviewer 3 Report
Accept in present form.